# Heat Treatment Augments Antigen Detection of *Dirofilaria immitis* in Apparently Healthy Companion Dogs (3.8% to 7.3%): Insights from a Large-Scale Nationwide Survey across the United States

**DOI:** 10.3390/pathogens13010056

**Published:** 2024-01-06

**Authors:** Daniel Felipe Barrantes Murillo, Annie Moye, Chengming Wang

**Affiliations:** 1Department of Pathobiology, College of Veterinary Medicine, Auburn University, Auburn, AL 36849, USA; dfb0014@auburn.edu; 2College of Sciences and Mathematics, Auburn University, Auburn, AL 36849, USA; acm0146@auburn.edu

**Keywords:** *Dirofilaria immitis*, heartworm, companion dogs, USA, antigen testing, heat treatment

## Abstract

Background: Heartworm disease (HWD) is a vector-borne disease caused by the filarial nematode *Dirofilaria immitis*. Low antigen levels caused by immune complex formation preclude HWD diagnosis. Heat treatment is an immune complex dissociation technique used to enhance antigen detection. Only a few studies have reported the benefits of heat treatment in nationwide surveys. Methods: To investigate the impact of heat treatment on the seroprevalence of HWD in companion dogs in the USA, serum samples (n = 3253) were analyzed for *D. immitis* antigen (DiroCHEK^®^, Zoetis) without and with heat treatment of the samples. Results: Compared to sera without heat treatment, heat treatment significantly increased overall prevalence from 3.8% (123/3253) to 7.3% (237/3253) (*p* < 10^−4^), expanding antigen detection from 32 to 39 of the 48 states and Washington District of Columbia included in this study. Conclusions: This study represents the largest nationwide survey of HW antigen detection in dogs in the US applying heat treatment to canine sera. The heat treatment used herein has the advantage of requiring a low volume of serum, making it optimal for use in routine diagnosis. Heat treatment should be used routinely by reference laboratories and veterinary clinics in patients with a negative initial test.

## 1. Introduction

Heartworm disease (HWD) is caused by the infection of the nematode *Dirofilaria immitis*, transmitted by several mosquito vectors from the genera *Aedes*, *Anopheles*, and *Culex* [1,2]. The microfilaria present in the blood is ingested by the mosquito vector during blood feeding, migrating to the Malpighian tubules to develop into an infective L3 larva [1]. After a mosquito’s blood meal, L3 infective larvae are inoculated within the subcutaneous space and migrate to the pulmonary arteries of the dog, becoming adult worms and initiating the production of microfilariae approximately 6–9 months after infection [1].

The presence of immature and adult worms induces damage to the pulmonary arteries and lungs by inciting eosinophilic inflammation and the release of toxic substances, causing endarteritis, villous proliferation, and endothelial damage [3]. Thrombosis and granulomatous inflammation are the most severe sequelae after the nematode’s death [3]. In cases of extreme parasite burden, animals develop deadly caval syndrome [3]. Most infected dogs do not exhibit clinical signs, regardless of the infection’s duration or the number of nematodes within the pulmonary arteries or right heart [3]. Other clinical signs are nonspecific and include weight loss, exercise intolerance, cough, and dyspnea, which can progress to pulmonary hypertension and congestive heart failure [3].

Diagnosis of HWD can be achieved by combining several diagnostic tests, including antigen detection by ELISA, microfilaria tests, PCR, and additional methods such as radiography and echocardiography [4,5]. Antigen testing is considered the gold standard for HWD, targeting a glycoprotein derived from the reproductive tract of female worms when it is free in the plasma or serum [3,6,7]. Antigen detection identifies a microfilaremic dog infected with at least one female worm [4]. However, antigen testing is not infallible and can yield false-negative results in patients with low levels of antigen, only male infections, or when the antigen is bound to excessive circulating antibodies [7].

Immune complex dissociation (ICD) techniques are used to unbind the antigen from antibodies by employing enzymes, applying acid reagents, or increasing the temperature of the samples [1]. Heat treatment before antigen testing has been demonstrated to increase HW detection in cases highly suspicious of HW infection [8,9]. Heat treatment is achieved by subjecting the sample to a temperature of 104 °C for 10 min; however, significant changes in the antigen binding to antibodies have been reported after applying temperatures above 65 °C [10,11,12,13,14,15,16,17,18,19,20]. The proposed mechanism of heat treatment is the denaturation of antibodies without affecting antigen stability and detection [9]. Heat treatment has also been documented to increase antigen detection only in male-infected dogs, confirmed by postmortem examination [12].

Despite being used since 1985 for heartworm diagnosis, ICD techniques have been removed from commercially available antigen tests due to the increased sensitivity of the test and higher quality of the reagents used [9,10,11] Heat treatment is not routinely performed in reference labs, and its use has been relegated to very specific conditions; for example, dogs from endemic regions without a known history of preventative use, dogs with an inconsistent prophylactic plan, microfilaremic animals with a previous negative antigen test, and infected patients recently treated with adulticides, preventatives, and doxycycline [8].

Since heat treatment is not routinely used for HW diagnosis in reference laboratories, we aimed to identify the impact of heat treatment ICD in a large-scale nationwide survey using serum samples from apparently healthy dogs in the US. We sought to provide additional information regarding the epidemiological distribution of *D. immitis* and report if there is any statistical difference between the prevalence rates before and after heat treatment. The heat treatment performed in this study was described by Weil et al. 1985 [21] and was chosen due to its demonstrated ability to retrieve antigens and the convenient low volume needed, as demonstrated in a previous study [22], since the serum available from each dog was limited. Previous studies described the impact of heat treatment before antigen testing in serum samples from shelter dogs in endemic areas [15,16,17,18,19,20]. To our knowledge, this is the largest nationwide serological survey of HW antigen detection using heat treatment in canine serum samples. We hypothesize that if heat treatment ICD is beneficial in reducing undetected cases, it could be considered for routine use in HW diagnosis in reference laboratories in combination with other diagnostic methods.

## 2. Materials and Methods

### 2.1. Serum Samples from Dogs

Serum samples from apparently healthy companion dogs (n = 3253) utilized in this study were submitted to Auburn University College of Veterinary Medicine between November 2016 and May 2022 for the determination of rabies antibody titers. The samples were submitted from 48 states of the continental USA and the Washington District of Columbia. Notably, no samples were available from South Dakota and North Dakota.

The serum samples had been previously heated at 56 °C for 30 min for fluorescent antibody virus neutralization (FAVN) and rapid fluorescent foci inhibition test (RFFIT) tests and were subsequently stored at −20 °C. For each sample taken, a 50 µL aliquot was directly used for antigen testing, and another 100 µL aliquot was used for heat treatment followed by antigen testing.

### 2.2. Heat Treatment and Antigen Testing

One aliquot from each serum sample was subjected to a heat treatment ICD described by Weil et al. 1985. Briefly, in a 1.5 mL microcentrifuge tube, 100 µL of serum and 100 µL 0.1 M disodium EDTA, pH 7.5, were mixed (1:1 ratio) and incubated in a heat block at 100 °C for 5 min. The tube and its contents were placed in a microcentrifuge and centrifuged at 16,000× *g* for 5 min [22].

For heartworm antigen detection, two aliquots of 50 µL of each sample, without and with heat treatment, were evaluated using a well-based commercial antigen capture ELISA (DiroCHEK^®^ Heartworm Antigen Test Kit Zoetis, Florham Park, NJ, USA) according to the manufacturer’s instructions. The interpretation of the antigen testing was conducted visually to determine the presence or absence of antigen by observing the color change on the DiroCHEK^®^ as indicated in the manufacturer’s instructions.

### 2.3. Statistical Analysis

The statistical analyses were conducted using the IBM^®^ SPSS^®^ Statistics software package 29.0 (International Business Machines Corp., Amonk, NY, USA). A McNear’s chi-square test was performed to compare the prevalence of *D. immitis* antigen detection before and after heat treatment, between dogs from different geographical regions, and between female and male dogs. A difference at *p* ≤ 0.05 was considered statistically significant.

## 3. Results

The overall HW antigen detection using DiroCHEK^®^ without heat treatment was 3.8% (123/3253) in dogs from 32 of the 48 states and Washington District of Columbia (32/47, 66.7%) (Table 1 and Appendix A, Figure 1). A statistically significant difference (chi-square test, *p* < 10^−4^) was observed in the total positive prevalence with heat treatment in sera, which resulted in a 7.3% antigen positive prevalence in 39 states of the 48 states and Washington District of Columbia (39/48, 81.3%) (Table 1 and Appendix A, Figure 1).

With heat treatment of the canine serum samples, 92.2% of the tested samples (3000/3253) remained negative, and 3.3% of samples (107/3253) remained positive (Table 2). Notably, 4.0% of samples that were initially negative (130/3253) converted to positive after the heat treatment. Interestingly, 16 of the initially positive samples (16/3253, 0.5%) turned negative with heat treatment (Table 2). The 16 samples were identified in Arkansas (n = 4), Arizona (n = 2), Massachusetts (n = 4), Minnesota (n = 2), New Hampshire (n = 1), Nevada (n = 9), Ohio (n = 4), and Oregon (n = 1) (Table 3). Only one of the four samples initially detected in Arkansas remained positive after heat treatment; none of the samples with initial detection of antigen from Arizona, New Hampshire, Ohio, and Oregon remained positive after heat treatment. The number of positives was reduced from four to three in Massachusetts, from two to one in Minnesota, and from nine to six in Nevada (Table 3).

The effect of heat treatment on the HW antigen detection rate did not show a significant difference between sera from female and male dogs (Table 2). In fact, a slightly higher percent of sera from female dogs converted from negative to positive with heat treatment (female: 72/1585, 4.5%; male: 57/1662, 3.4%; *p* = 0.10) (Table 2).

The samples were sorted by geographical location into four regions based on the CDC’s geographic region definitions (https://www.cdc.gov/nchs/hus/sources-definitions/geographic-region.htm accessed on 15 November 2023). Heat treatment of the sera also led to significantly higher prevalences in dogs from the South, Northeast, and Midwest regions (Table 1). Antigen detection prevalence in the South region of the USA (1.6%, 51/1455) was the highest, followed by the West (1.4%, 44/833), Northeast (0.6%, 19/566), and the Midwest (0.3%, 9/399) (Table 1).

The difference between the mean ages of antigen-positive (6.0 ± 9.8 years) and antigen-negative dogs (5.81 ± 8.8) was not statistically significant (*p* = 0.71). The mean age of the positive dogs after heat treatment was 5.2 years (±4.0 standard deviation) with a range from 4 months to 19 years old. There was no statistically significant difference in the prevalence of HW antigen detection by sex (female: 8.1%, 128/1585; male: 6.5%, 108/1662; *p* = 0.08). The distribution of the cases per year was 3.1% (22/716) in 2022, 8.4% (115/1367) in 2021, 10.2% (57/561) in 2020, 4.9% (14/284) in 2019, 3.4% (6/175) in 2018, and 15.6% (23/147) in 2017 (Appendix A).

After heat treatment, positive sample distribution by breed was the following: mixed breeds and crossbreeds were the most frequent samples (n = 66), followed by Yorkshire Terriers, German Shepherds, and Golden Retrievers, with nine individuals each (Appendix A). Australian Shepherds and Labrador Retrievers were represented by eight individuals each. The next most represented breeds were Poodle (6), Pembroke Welsh Corgi, Maltese, and Dachshund, with five individuals each, and Shih Tzu, Pomeranian, Miniature Schnauzer, and Bichon Frise, with four individuals each. The remaining breeds were represented by three individuals or fewer (Appendix A).

## 4. Discussion

In this study, the overall nationwide prevalence of HW antigen without heat treatment was 3.8%. This is much higher than the previously reported nationwide heartworm prevalence. A prior study documented an overall prevalence of heartworm antigen detection by a commercial SNAP test of 1.4% between 2001 and 2007 [1]. Subsequent nationwide surveys assessing antigen detection from commercial ELISA kits used in veterinary clinics and reference laboratories reported rates of 1.3% between 2010 and 2012 and 1.4% between 2013 and 2019 [23,24]. For instance, the CAPC reported an antigen seroprevalence of HW ranging from 1.22% to 1.34% from 2016 to 2022 (Canine Animal Parasite Council, CAPC) [25]. Our data reflect an almost twofold difference when compared with the literature. However, this discrepancy has been documented previously. A previous nationwide survey in cats yielded conflicting results when comparing antibody prevalence versus CAPC data [25,26].

The difference between the antigen positive rates from the prior reports and this study can be a consequence of several factors. The data sources can have some limitations; for example, CAPC prevalence maps are constructed using data from IDEXX Laboratories, Antech Diagnostics, and Zoetis Diagnostics, but they are estimated to represent less than 30% of the real infection rates in the different regions of the US (CAPC). Also, we speculate that HWD is widely underdiagnosed, as not every dog undergoes annual testing for *D. immitis* infection. Antigen testing is considered the most sensitive method for HW diagnosis in animals without clinical signs or suspicious cases of infection and is recommended to be performed annually [27]. It is possible that the number of dogs on monthly preventatives or being screened annually for HW antigen is lower than estimated. Our sample pool comprises apparently healthy animals without clinical signs, and the samples were submitted for rabies antibody testing before international or domestic travel. Many dogs can be infected with *D. immitis* and show no clinical signs, especially if their level of physical activity is low [27]. Mild and moderate HW disease has several clinical signs that are more conspicuous and incited by physical activity, such as coughing, abnormal lung sounds, and exercise intolerance [27]. Not only can exercise affect the severity of the clinical signs in infected dogs, but also the number of worms relative to the size of the dog, the host immune response, and the chronicity of infection [27]. Our data reflect that the population screened is not “apparently healthy dogs” but animals with subclinical infection.

Geographical distribution revealed a statistically significant difference among the regions of the US (chi-square test, *p* < 0.0001) before heat treatment, with the South region having the highest prevalence (1.6%). This finding is in accordance with previous publications that identified the South region of the US with higher HW prevalence [1,23,24]. However, the Northeast region was reported to have the lowest prevalence rates of HW detection [1,23,24], unlike in our study, in which the region with the lowest prevalence was the Midwest (0.3%). This discrepancy with the literature is explained by the uneven distribution of the samples among the states, leading to marked differences in the available samples from each region. For example, we retrieved 198 samples from the state of Pennsylvania. Similarly, the yearly distribution of cases was statistically significant (chi-square test, *p* ≤ 0.001) due to the uneven distribution per year (only in 2021, a total of 1367 samples were retrieved).

Antigen testing should not be performed alone, without other diagnostic techniques. Specifically, microfilaria testing is recommended to be performed along with antigen testing since the antigens can be blocked by circulating antibodies, and in some instances, infected dogs can even be antigen-negative without microfilaremia [27]. To better understand the impact of antigen–antibody complexes in HW diagnosis, we took two aliquots from each sample (without and with heat treatment) for antigen detection in this study. The heat treatment elected can retrieve HW antigens after serial dilutions and only needs a small volume per sample, 100 µL [21,22]. The low volume needed makes this technique extremely convenient, especially when serum is limited in experimental assays. After heat treatment, the prevalence increased to 7.3% with a significant statistical difference (chi-square test, *p* < 0.0001). A previous study performing evaluation antigen detection using a commercially bound ELISA (SNAP^®^ 4Dx^®^ Test, IDEXX Laboratories Inc., Westbrook, ME, USA) on 165 HW negative plasma samples from dogs from animal shelters across the US reported an increase in prevalence of 7.1% after heat treatment [15]. Heat treatment could only be performed in 154 of the negative samples due to the limited amount of plasma; the protocol used required 0.7–1.5 mL per sample, thus limiting the number of samples processed [15]. 

Another study evaluated the use of heat treatment ICD before antigen testing by using DiroCHEK^®^ Heartworm Antigen Test Kit (Zoetis, Florham Park, NJ, USA) in 616 blood samples from shelter dogs in Florida, Louisiana, and Texas (South); Ohio and New York (North); and Colorado and California (West) [20] (DiGangi et al., 2017). A modified Knott’s test was performed on each sample, and a total of 74 dogs were detected with HW antigen, 29 (39.2%) of which after heat treatment, increasing the prevalence rate from 7.3% to 12.3% [20]. Heat treatment could not be evaluated in all the samples (n = 558) due to insufficient volume after heating, solidification, or hemolysis [20]. The heat treatment consisted of mixing the negative samples in 0.9% saline to minimize coagulation [20]. The authors also determined risk factors for antigen blocking in negative samples before heat treatment. Such factors include the presence of microfilaremia, history of preventive administration, and geographical region [20]. It is documented that microfilaremic animals without antigen detection are likely to convert to positive after heat treatment [8]. The use of preventatives intermittently will lead to the worm’s death, causing inflammatory changes and antigen sequester by antibodies [8].

Our results align with these previous surveys, with an increased prevalence of HW antigen detection after heat treatment; however, the differences of the reported values (7.3% vs. 7.1% and 12.3%, respectively) are related to the population screened. Dogs included in previous studies come from animal shelters, which are considered to have an increased risk of HWD [15,16,17,18,19,20]. The distribution of our samples was broader, across 48 states and Washington District of Columbia, and all the dogs included were assumed to be home-owned since they were subjected to rabies testing due to previous travel. The number of samples (n = 3253) was significantly higher than those in previous studies (n = 165 and n = 616), making this survey more representative. Limitations on the serum available were reported in previous studies using heat treatment [15,20]. This limitation was not experienced in our study, since heat treatment was performed in all the available samples retrieved (n = 3253). 

There was not a statistically significant difference between the ages of the antigen-positive animals (chi-square test, *p* = 0.71). The mean age of the antigen-positive dogs after heat treatment was 5.22 years (±4.18 standard deviation). Interestingly, after heat treatment, animals as young as 4 months were detected positive in this study. Previous studies do not include animals younger than 6 months [20]. However, a study with dogs experimentally infected by *D. immitis* demonstrated that antigen detection can be achieved after heat treatment as early as 98 days (126.9 mean, ±18.9 SD) using DiroCHEK^®^ and 105 days (131.5 mean, ±11.7 SD) using PetChek^®^ (IDEXX Laboratories Inc.) [28].

Another interesting finding were the samples that initially tested positive through the antigen test but converted to negative after heat treatment (n = 16). These samples were identified in Arkansas (n = 4), Arizona (n = 2), Massachusetts (n = 4), Minnesota (n = 2), New Hampshire (n = 1), Nevada (n = 9), Ohio (n = 4), and Oregon (n = 1), representing 0.5% of the total samples tested. False-positive results are reported in wildlife species infected with other nematodes belonging to the *Dirofilaria* genus and in samples without heat treatment [6,8,29,30,31,32]. Heat treatment is unlikely to induce cross-reactions, according to one study which used positive samples from dogs confirmed by necropsy [9]. Moreover, another study stated the opposite, and concluded that heat treatment of initially negative blood samples for *D. immitis* could lead to false positive antigen test results if the dog is infected with *Acanthocheilonema dracunculoides* [33].

Our study has several limitations that are important to mention. The history of the dogs used in this project is limited to the clinical features of each animal, including breed, age, sex, and geographical location. No further details regarding lifestyle, physical activity, previous traveling history, preventative use, or heartworm status are mentioned, precluding a more detailed analysis. Only serum was available from each dog, without whole blood; thus, microfilaria tests or PCR could not be performed. The use of microfilaria tests and antigen detection kits is recommended when screening healthy animals or animals with a high index suspicion for HW infection [27]. A total of 0.5% percent of the samples tested was determined as possible “false positives” that became negative after heat treatment in this project; unfortunately, a gold standard for HWD diagnosis is lacking, and thus this could not be determined. Ideally, infected dogs should be confirmed through the demonstration of adult worms in the right heart or pulmonary arteries by necropsy and histopathology analysis. Concurrent infections with other parasites from the genus *Dirofilaria* or intestinal and subcutaneous nematodes could not be determined due to the limited information available from each dog; thus, possible cross-reactions causing false positives in this study cannot be discussed. Nevertheless, this nationwide survey, which involved the heat treatment of canine sera, strongly suggests that the true prevalence of *D. immitis* in dogs in the USA could be significantly higher than previously documented.

## 5. Conclusions

In conclusion, this study represents the largest nationwide survey of HW antigen detection in dogs in the US, applying heat treatment to 3253 canine sera. Heat treatment significantly increased the prevalence from 3.8% (123/3253) to 7.3% (237/3253). Our results sharply contrast with the CAPC data from those years (2016–2022), in which prevalence ranges from 1.22% to 1.34%, revealing that HWD status is widely underestimated in companion dogs in the US. The heat treatment used herein has the advantage of requiring a low volume of serum or plasma, making it optimal for routine diagnosis. We advocate for the routine use of heat treatment by reference laboratories and veterinary clinics in patients with a negative initial test. Additionally, microfilaria testing should be employed alongside antigen detection to enhance diagnostic accuracy. Further research and widespread adoption of heat treatment in routine heartworm diagnostics are warranted to address the underdiagnosis of heartworm disease in companion dogs and improve overall management strategies.

## Figures and Tables

**Figure 1 pathogens-13-00056-f001:**
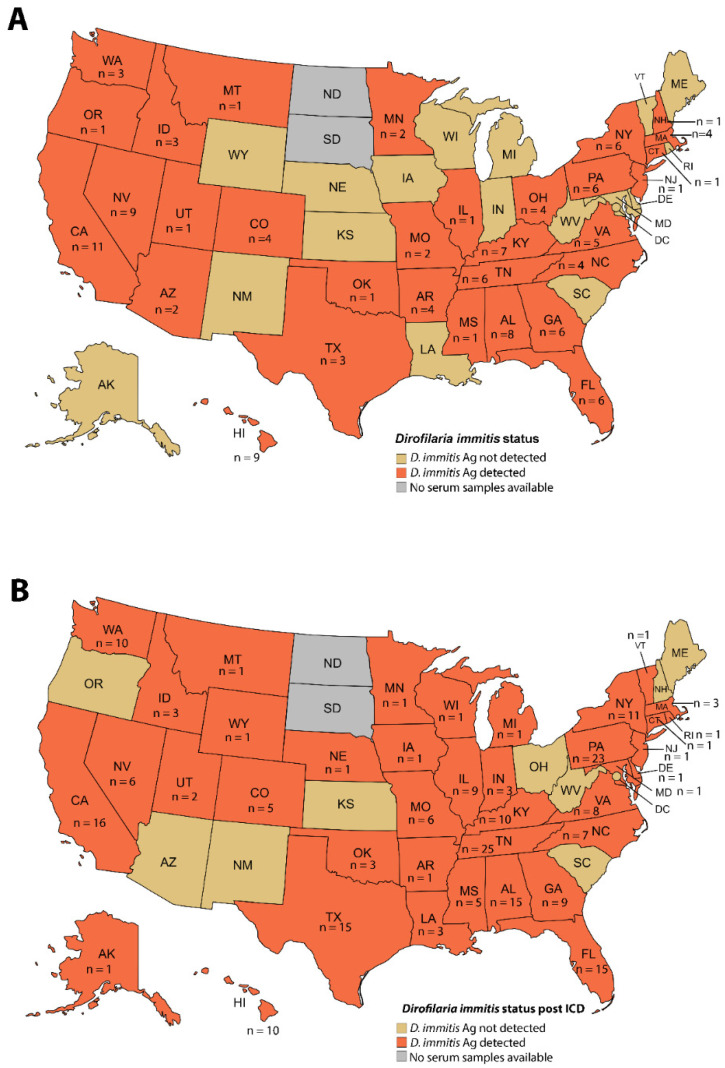
Distribution of HW antigen positivity by state before (**A**) and after (**B**) ICD with heat treatment.

**Table 1 pathogens-13-00056-t001:** Rate of HW antigen detection in canine serum samples of dogs from different regions without and with heat treatment.

Region * (N Animals Tested)	Without Heat Treatment (N Positive; %)	With Heat Treatment (N Positive; %)	*p* Value
South (1455)	51/1455; 3.5%	118/1455; 8.1%	*p* < 0.0001
West (833)	44/833; 5.3%	55/833; 6.6%	*p* = 0.25
Northeast (566)	19/566; 3.4%	41/566; 7.2%	*p* = 0.004
Midwest (399)	9/399; 2.3%	23/399; 5.8%	*p* = 0.011
Total (3253)	123/3253; 3.8%	237/3253; 7.3%	*p* < 0.0001

* The definition of the different regions can be found at https://www.cdc.gov/nchs/hus/sources-definitions/geographic-region.htm (accessed on 16 December 2023).

**Table 2 pathogens-13-00056-t002:** Change of HW antigen detection in serum samples without and with heat treatment.

Positivity of HW Antigen Detection Post Heat Treatment	Female	Male	Unknown Sex	Total
Remained negative	1448	1547	5	3000
Remained positive	56	51	0	107
Turned to positive	72	57	1	130
Turned to negative	9	7	0	16
Total	1585	1662	6	3253

**Table 3 pathogens-13-00056-t003:** HW antigen detection prevalence in dog sera without and with heat treatment.

State * (Number of Samples Tested)	Without Heat Treatment (N Positive; %)	With Heat Treatment (N Positive; %)
Alaska (7)	0, 0.0%	1, 14.3%
Alabama (192)	8, 4.2%	15, 7.8%
Arkansas (21)	4, 19.0%	1, 4.8%
Arizona (39)	2, 5.1%	0, 0.0%
California (193)	11, 5.4%	16, 8.3%
Colorado (91)	4, 4.4%	5, 5.5%
Connecticut (24)	1, 4.2%	1, 4.2%
Delaware (9)	0, 0.0%	1, 11.1%
Florida (190)	6, 3.2%	15, 7.9%
Georgia (196)	6, 3.1%	9, 4.6%
Hawaii (198)	9, 4.5%	10, 5.1%
Iowa (12)	0, 0.0%	1, 8.3%
Idaho (36)	3, 8.3%	3, 8.3%
Illinois (81)	1, 1.2%	9, 11.1%
Indiana (36)	0, 0.0%	3, 8.3%
Kentucky (141)	7, 5.0%	10, 7.1%
Louisiana (46)	0, 0.0%	3, 6.5%
Massachusetts (99)	4, 4.0%	3, 3.0%
Maryland (22)	0, 0.0%	1, 4.5%
Michigan (49)	0, 0.0%	1, 2.0%
Minnesota (29)	2, 6.9%	1, 3.4%
Missouri (47)	2, 4.3%	6, 12.8%
Mississippi (30)	1, 3.3%	5, 16.7%
Montana (8)	1, 12.5%	1, 12.5%
North Carolina (74)	4, 5.4%	7, 9.5%
Nebraska (6)	0, 0.0%	1, 16.7%
New Hampshire (6)	1, 16.7%	0, 0.0%
New Jersey (30)	1, 3.3%	1, 3.3%
Nevada (103)	9, 8.7%	6, 5.8%
New York (193)	6, 3.1%	11, 5.7%
Ohio (117)	4, 3.4%	0, 0.0%
Oklahoma (30)	1, 3.3%	3, 10.0%
Oregon (21)	1, 4.8%	0.0%
Pennsylvania (198)	6, 3.0%	23, 11.6%
Rhode Island (3)	0, 0.0%	1, 33.3%
Tennessee (197)	6, 3.1%	25, 12.9%
Texas (163)	3, 1.8%	15, 9.2%
Utah (27)	1, 3.7%	2, 7.4%
Virginia (119)	5, 4.2%	8, 6.7%
Vermont (8)	0, 0.0%	1, 12.5%
Washington (99)	3, 3.0%	10, 10.1%
Wisconsin (16)	0, 0.0%	1, 6.3%
Wyoming (4)	0, 0.0%	1, 25.0%
Total (3253)	123/3253, 3.8%	237/3253, 7.3%

* The HW antigen was not detected in dog sera from the following states: Kansas (n = 6), Maine (n = 5), New Mexico (n = 7), South Carolina (n = 16), West Virginia (n = 10), and Washington District of Columbia (n = 2).

## Data Availability

The datasets generated during the current study are available from the corresponding author on reasonable request.

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
