# Peer review of "Heat Treatment Augments Antigen Detection of *Dirofilaria immitis* in Apparently Healthy Companion Dogs (3.8% to 7.3%): Insights from a Large-Scale Nationwide Survey across the United States"

_pathogens, 2024, doi:10.3390/pathogens13010056_

Round 1

Reviewer 1 Report

Comments and Suggestions for Authors

Main concern is related to the samples as NAD after heat treatment but positive before heat treatment.  That would indicate that current tests are providing false positive results.  It is recommended that the statements about other Dirofilaria sp. infections in the paper can be backed up with more information.

Author Response

Dear Reviewer,

Please find the responses to your comments in the attached PDF file.

Happy new Year!

Chengming

Reviewer 2 Report

Comments and Suggestions for Authors

OVERALL COMMENTS

 This study aimed to evaluate Dirofilaria immitis prevalence in the USA by the use of an antigen test with and without immune complex dissociation (ICD) by heat treatment across a population of n = 3,253 “healthy” dogs. The authors used the difference between the overall two prevalence values, 3.8% and 7.3% without and with heat treatment, respectively, to highlight the importance of including the ICD approach for routine screening analyses, in association with other HW diagnostic methods. As the authors suggested, this will also allow the detection of subclinical cases (“apparently healthy dogs”) otherwise diagnosed as negative. In addition, this study has been one of the largest surveys for D. immitis detection in the USA, reporting also the highest prevalence data ever recorded in this geographical area. The manuscript is well written and results and conclusions are relevant. My only concern is with the result section, for which the flow can be improved and the text summerised to avoid redundancy with what is reported in the tables. Therefore, in my opinion, the manuscript is suitable for publication in Pathogens, pending minor revisions.

MINOR COMMENTS

Results

Lines 130 and 133: The prevalence values of 66.7% and 81.3% here reported are not very clear. I understand that they refer to the specific prevalence in the Washington District of Columbia, but it would be clearer if the number of positive dogs/ total animal tested could also be reported in brackets.

Lines 144-148: This part of the paragraph should be reworded and summarised, as the main results are already showed in table 2 in detail.

Line 163:  Replace “cats” with “dogs”.

Lines 178-179: These two sentences should be linked by a column.

Lines 185- 186: This sentence can be moved above when discussing the significance of the mean age and HW antigen positivity (line 172). Also, I understand the authors’ intent to evaluate the average age among positive and negative dogs and compare the two values. However, in my opinion, it would be more relevant if the authors also calculate the correlation between age and HW positivity, by dividing the studied population in age groups (e.g., youngs, adults, old) and calculating the statistical significance for each group.

Table 1: I would suggest to slightly modify the table in order to be easier to understand by the reader, avoiding replicates of same values. For instance:

        Region*

(N animal tested)

Without heat treatment

             N positive (%)

With heat treatment

             N positive (%)

P value

South (1,455)

51 (3.5)

118 (8.1)

<0.0001

West (833)

44 (5.3)

55 (6.6)

0.25

Northeast (566)

19 (3.4)

41 (7.2)

0.004

Midwest (399)

9 (missing perc)

23 (5.8)

0.011

Total (3,253)

123 (3.8)

237 (7.3)

<0.0001

Figure 1: I believe the caption is too long and redundant. I would modify as follows: “Distribution of HW antigen positivity by state before (A) and after (B) ICD with heat treatment”.

Table 2: Please, change: “gender” in “sex”; “from without heat treatment to with heat treatment” in “post heat treatment”; “from negative to positve” in “turned positive”; “from positive to negative” in “turned negative”.

Table 3: I would suggest reporting the prevalence percentage values in brackets as showed in table 1 (see previous comment). Percentage values should also be showed for the total (last line).

Discussion

Line 195: Please include full name and reference to the website for CAPC.

Line 295: I would suggest rewording the beginning of this sentence as follows: “Heat treatment is unlikely to induce cross-reactions ... ”. Moreover, in light of another study stating the opposite (Szatmári, V., van Leeuwen, M.W., Piek, C.J. et al. False positive antigen test for Dirofilaria immitis after heat treatment of the blood sample in a microfilaremic dog infected with Acanthocheilonema dracunculoides. Parasites Vectors 13, 501 (2020). https://doi.org/10.1186/s13071-020-04376-9).

Lines 305-313: Considering the previous HW prevalence data available, how would the authors expect the positive/negative predicted values for the HWD could potentially affect the results obtained in this study? It would be interesting to discuss this point, also in light of the false positive results obtained/ sera turning negative at the heat treatment and for stressing the point that the antigen test alone is not enough for the HW diagnosis.

Author Response

(The authors gave the same response as above.)
